# Infective Endocarditis in Patients with End-Stage Renal Disease on Dialysis: Epidemiology, Risk Factors, Diagnostic Challenges, and Management Approaches

**DOI:** 10.3390/healthcare12161631

**Published:** 2024-08-16

**Authors:** Rochell Issa, Nourhan Chaaban, Abdullah Salahie, Bianca Honnekeri, Gary Parizher, Bo Xu

**Affiliations:** 1Department of Internal Medicine, Cleveland Clinic, Cleveland, OH 44195, USA; 2Department of Cardiology, University of Toledo, Toledo, OH 43606, USA; 3Section of Cardiovascular Imaging, Robert and Suzanne Tomsich Department of Cardiovascular Medicine, Sydell and Arnold Family Heart, Vascular, and Thoracic Institute, Cleveland Clinic Foundation, Cleveland, OH 44195, USA

**Keywords:** infective endocarditis, end-stage renal disease, dialysis, cardiovascular imaging

## Abstract

Infective endocarditis (IE) poses a significant clinical challenge, especially among patients with end-stage renal disease (ESRD) undergoing dialysis, and is associated with high morbidity and mortality rates. This review provides a contemporary discussion of the epidemiology, risk factors, diagnostic challenges, and management strategies for IE among ESRD patients, including a literature review of recent studies focused on this vulnerable population. The review highlights the multifactorial nature of IE risk in ESRD patients, emphasizing the roles of vascular access type, dialysis modality, and comorbid conditions. It also explores the diagnostic utility of different imaging modalities and the importance of a multidisciplinary approach in managing IE, including both medical and surgical interventions. The insights from this review aim to contribute to the improvement of patient outcomes through early recognition, appropriate antimicrobial therapy, and timely surgical intervention when necessary.

## 1. Introduction

Infective endocarditis (IE) in patients with end-stage renal disease (ESRD) on dialysis poses a significant clinical challenge and is a significant cause of morbidity and mortality, with mortality rates ranging between 20 and 40% despite advancements in diagnostic techniques and management [1,2]. The incidence of IE among dialysis patients, particularly those on hemodialysis, is significantly higher compared to patients not on dialysis and has been increasing in recent years. According to a study from the Nationwide Inpatient Sample database between 2006 and 2011, the hospitalization rate of bacterial endocarditis in dialysis patients increased from 175 to 222 per 10,000 patients within the United States, indicating a rising trend in the incidence of IE among dialysis patients [3]. In comparison, the incidence of IE in the general population is significantly lower, around 0.18 per 1000 persons per year, with an estimated incidence rate ratio of 38.1 [4]. Several factors contribute to this elevated incidence of IE in ESRD patients on dialysis, including frequent vascular access device use, immune system dysfunction, and higher exposure to hospital-acquired infections [5]. Clinical manifestations of IE present in dialysis patients include fever, chills, and signs of systemic infections, as seen in non-dialysis patients [6]. Moreover, underlying conditions such as diabetes mellitus and hypertension further elevate IE risk for dialysis patients [7]. The interaction of these factors highlights the complex pathophysiology of IE in dialysis patients, emphasizing the need for an in-depth understanding to guide effective diagnostic and treatment approaches [8]. Timely and accurate diagnosis, coupled with antimicrobial therapy and surgical intervention when clinically indicated, are pivotal in enhancing outcomes and lowering mortality rates in this high-risk population (Figure 1). The goal of this review is to highlight the multifactorial nature of IE risk in dialysis patients, emphasize the roles of vascular access type, dialysis modality, and comorbid conditions in this vulnerable patient population, and explore the diagnostic utility of different imaging modalities and the importance of a multidisciplinary approach in managing IE, including both medical and surgical interventions, while providing a focused literature review of the most current prospective/retrospective studies in the last 15 years. The insights from this review aim to contribute to the improvement of patient outcomes through early recognition, appropriate antimicrobial therapy, and timely surgical intervention when necessary.

## 2. Risk Factors

ESRD patients undergoing dialysis face a significantly heightened risk of developing IE compared to the general population [9]. This increased susceptibility is multifactorial and influenced by various patient-related and treatment-related factors.

### 2.1. Uremia-Induced Immune Dysfunction

Chronic kidney disease (CKD) and ESRD are associated with immune system dysfunction, characterized by impaired phagocytic activity, reduced neutrophil chemotaxis, and dysfunctional leukocyte responses [10]. Uremia-induced immune dysfunction compromises the host’s ability to effectively clear infective pathogens, predisposing ESRD patients to infections, including IE [11].

### 2.2. Dialysis Modality

The choice of dialysis method can impact the likelihood of IE in patients with ESRD. Hemodialysis (HD) and peritoneal dialysis (PD) are the two methods utilized for ESRD patients. HD has been associated with a higher incidence of IE as compared to PD [12]. This is partly due to the use of access devices such as to the repeated use of vascular access devices, such as arteriovenous fistulas (AVFs), arteriovenous grafts (AVGs), and central venous catheters (CVCs), leading to a greater risk of bacteremia in HD patients than in PD patients [13]. Nonetheless, PD patients may still develop IE, although it occurs frequently primarily due to bacteremia triggered by peritonitis within this patient group [14,15]. In our analysis presented in Table 1, we observed that HD was largely the dialysis modality used among ESRD patients diagnosed with IE.

### 2.3. Vascular Access Type

The type of vascular access used for hemodialysis is another significant risk factor for IE [16]. AVFs are considered the preferred access due to their survivability and lower risk of infection compared to arteriovenous grafts (AVGs), and especially as compared to CVCs [16]. This was consistent within our review, as AVF usage ranged from 14.7% (Zolfaghari et al.) to 69% (Durante-Mangoni et al.), highlighting its role as a preferred access type due to lower infection rates (Table 1) [17,18]. However, despite their lower infection rates long term, AVFs are not without risk. More specifically, these patients have an even higher rate of catheter-related bacteremia prior to AVF maturation [19]. This is likely why AVFs have been associated with 34.2% of IE cases in HD patients [2]. However, it is difficult to ascertain whether this rate of IE is due to a lack of AVF maturation as much of the available literature does not disclose the age of the AVFs (Table 1).

**Table 1 healthcare-12-01631-t001:** Characteristics and risk factors of dialysis patients with infective endocarditis (IE). This table provides a comprehensive overview of the 8 reviewed studies, conducted between 2010 and 2024, assessing IE in dialysis patients. The table includes demographic information, inflammatory markers, dialysis types, vascular access methods, and associated comorbidities. Abbreviations: hemodialysis (HD); Erythrocyte Sedimentation Rate (ESR); C-reactive protein (CRP); non-hemodialysis (non-HD); arteriovenous fistula (AVF); central venous catheter (CVC); hypertension (HTN); diabetes mellitus (DM); congestive heart failure (CHF); arteriovenous graft (AVG); peripheral vascular disease (PVD); cerebrovascular disease (CVD); myocardial infarction (MI); Cardiovascular Implantable Electronic Devices (CIED); intravenous drug use (IVDU); peritoneal dialysis (PD); systemic lupus erythematous (SLE).

Study/Country	Mean Age (Years)	Sex	Dialysis Patients with IE	Inflammatory Markers	Dialysis Type	Dialysis Access	Risk Factors vs. Non-Dialysis Patients
Zolfaghari et al., 2024 [17] Iran	49.8	Male: 23/34 (67.6%) Female: 11/34 (32.3%)	34	HD: ESR 54.4, CRP 59.1 Non-HD: ESR 49.8, CRP 48.5	HD	AVF: 5/34 (14.7%) CVC: 27/34 (79.4%) Both: 2/34 (5.8%)	HTN: 21/34 (61.76%) DM: 13/34 (38.23%) CHF: 10/34 (29.41%)
Pericas et al., 2021 [2] International	59.9	Male: 228/553 (41.4%) Female: 323/553 (58.6%)	553	Not described	HD	AVF: 189 (34.2%) AVG: 12 (2.5%) CVC: 235 (48.1%)	DM: 226/549 (41.2%) CHF: 77/195 (39.5%) Ischemic Heart Disease: 37/266 (13.9%) PVD: 58/266 (21.8%) CVD: 38/272 (14%)
Gallacher et al., 2021 [20] Scotland	58.9	Male: 123 (56.9%) Female: 93 (43.1%)	216	Not described	Not specified (HD, PD)	Not Described	CHF: 24 (11.1%) CVD: 16 (7.4%) MI: 14 (6.5%) Cardiac Device: 9 (4.2%) Prior Valve Surgery: 18 (8.3%)
Kwon et al., 2021 [13] Korea	64.4	Male: 15/34 (44.1%) Female: 19/34 (55.8%)	34	HD: ESR 92, CRP 15.41 Non-HD: ESR 88.5, CRP 10.61	HD	AVF: 9/34 (26.4%) AVG: 16/34 (47%) CVC: 9/34 (26.4%)	HTN: 26/34 (76.47%) DM: 15/34 (44.1%) CHF: 6/34 (17.6%) MI: 3/34 (8.8%) CVD: 5/34 (14.7%)
Bhatia et al., 2017 [3] United States	59.3	Male: 23,877/44,816 (53.3%) Female: 20,939/44,816 (46.7%)	44,816	Not described	HD and PD	Not Described	DM: 9677/44,816 (21.6%) CHF: 9559/44,816 (21.3%) Valvular Disease: 5858/44,816 (13.1%) CIEDs: 4907/44,816 (10.9%) Prior Valve Repair/Replacement: 1.908/44,816 (4.3%) IVDU:1863/44,816 (4.2%)
Durante-Mangoni et al., 2016 [18] Italy	66	Male: 29/42 (69%) Female: 13/42 (31%)	42	HD: 36/42 elevated CRP (86%), 27/42 elevated ESR (64%) Non-HD: 116/126 elevated CRP (93%), 86/126 elevated ESR (68%)	HD	AVF: 29/42 (69%) AVG:0/42 (0%) CDC: 13/42 (31%)	DM: 18/42 (42.9%) CHF: 17/42 (40.5%) CVD: 6/42 (15%) Congenital Heart Disease: 2/42 (4.8%) IVDU: 5/42 (11.9%) Prior IE History: 3/42 (7.1%) Cancer: 7/42 (16.7%) Immunosuppressive Treatment: 7/42 (16.7%)
Jones et al., 2013 [21] United Kingdom	55.2	Male: 22/42 (55.2%) Female: 20/42 (47.8%)	42	Not described	HD: 40/42 (95%) PD: 2/42 (5%)	Vascular Catheter: 4/42 (10%) Tunneled Line: 22/42 (55%) AVF: 14/42 (35%) PD: 2/42 (4.8%)	HTN: 28/42 (66.6%) DM: 14/42 (33.3%) Valvular Disease: 9/42 (21.4%) Immunosuppressive Regimen: 4/42 (9.5%) Heart and Lung Transplant: 1/42 (2.4%)
Tao et al., 2010 [22] China	52.3	Male: 4/6 (66.6%) Female: 2/6 (33.3%)	6	Not described	HD: 6/6 (100%)	Permanent Catheter: 3/6 (50%) Temporary Catheter: 2/6 (33.3%) AVF: 1/6 (16.6%)	HTN: 1/6 (16.6%) SLE: 1/6 (16.6%) Chronic Glomerulonephritis: 3/6 (50%)

AVGs and CVCs, on the other hand, have higher rates of infection due to their artificial nature and increased susceptibility to biofilm formation [23]. CVCs, in particular, are associated with the highest risk of bloodstream infections, making them a significant risk factor for IE for patients on dialysis [23]. Catheter-related infections can lead to local venous endothelial damage and facilitate the seeding of infective pathogens onto cardiac valves, contributing to the development of IE [24]. Despite this, our review showed high usage in many of the cohorts due to the ease of access, despite higher infection risks. For example, the Zolfaghari et al. study had a 79.4% usage while the Pericas et al. study had a 48.1% usage of CVCs among HD patients. Furthermore, patients with CVCs have a two to three times higher rate of mortality secondary to infection compared to patients with AVFs [25]. This may indicate that CVCs are being overused in this population or that they are not being promptly removed after initiation of dialysis, also seen within our review as some patients in the Tao et al. study as well as the Jones et al. study had both AVFs and CDCs [21,22]. Alternatively, AVGs have been associated with anywhere between 2.5% of IE cases in HD patients, such as with the Pericas et al. study, to 47% in the Kwon et al. study [2,13]. Overall, AVFs, AVGs, and CVCs are all associated with an increased risk of bacteremia, which is likely due to recurrent nosocomial vascular access, contributing to the higher IE rates in HD patients than in PD patients.

### 2.4. Comorbid Conditions

Patients undergoing dialysis often have a range of health conditions that can increase their risk of IE. For example, the rates of diabetes mellitus, hypertension, and cardiovascular disease are often higher in patients on HD compared to those not on dialysis, as shown in Table 1 [26]. Additionally, ischemic heart disease, PVD, cerebrovascular disease, and congestive heart failure have all been shown to be prevalent at higher rates in HD patients than in non-HD patients with IE, as can be seen in Table 1 [2]. These comorbidities contribute to endothelial damage, immune dysfunction, and systemic inflammation, creating an environment that makes it easier for microorganisms to infect cardiac valves and lead to IE [27].

### 2.5. Healthcare Exposures

Dialysis patients frequently require medical interventions and hospitalizations, exposing them to healthcare-associated infections. Risk factors for nosocomial infections in dialysis patients include longer durations on dialysis, longer dialysis sessions, an A1c > 7, and more frequent healthcare encounters, many of which were not looked at specifically within the studies documented in Table 1 [28,29]. These nosocomial exposures increase the risk of acquiring infective pathogens implicated in IE, such as Staphylococcus aureus and Enterococcus species, contributing to the heightened IE risk in this population [30]. In fact, one prospective cohort study found that 19.9% and 60.2% of IE cases in HD patients were nosocomial and non-nosocomial healthcare-acquired infections [2].

Understanding these multifactorial risk factors specific to patients on dialysis is crucial for the early recognition and effective management of IE. Comprehensive strategies aimed at mitigating these risk factors, including more standardized vascular access management, dialysis parameters, and infection control practices, are essential for reducing the incidence and morbidity associated with IE in this patient population.

## 3. Common Pathogens in Infective Endocarditis Dialysis Patients

In patients undergoing dialysis, IE is often caused by a variety of pathogens, including bacteria and occasionally fungi. The types of organisms responsible for IE can differ among patients receiving dialysis according to the type of dialysis and the associated risk of bloodstream infections. HD patients, for instance, have a likelihood of exposure to healthcare environments, and are at an increased risk of healthcare-associated infections due to their vascular access points, particularly CVCs [31,32]. Staphylococcus species infections, including Methicillin-resistant Staphylococcus aureus (MRSA), Methicillin-Sensitive Staphylococcus aureus (MSSA), and coagulase-negative Staphylococcus (i.e., Staphylococcus epidermidis), account for around 70% of IE cases in HD patients [33]. This aligns with the findings from our study, as presented in Table 2.

The risk of *S. aureus* infection is generally higher in patients with CVCs than in those with AVFs or AVGs because CVCs provide a direct route for bacteria to enter the bloodstream [34]. Other common causes of IE in both HD and non-HD patients include HACEK organisms, non-HACEK Gram-negative bacteria, polymicrobial infections, and microorganisms that do not show up on blood cultures [35]. Furthermore, fungal infections such as those caused by Candidal species are more common among this group due to the use of vascular access, although they are less prevalent than bacterial infections (e.g., accounting for 8.8% of causative agents according to Zolfaghari et al.) [3,17]. Apart from pathogens associated with healthcare settings, species like Streptococcus viridians and Streptococcus bovis are also frequently linked to IE among patients with ESRD undergoing dialysis, albeit at lower rates compared to non-dialysis patients as outlined Table 2 [2].

Conversely, PD patients face a higher risk of peritoneal infections related to their catheter, leading to a distinct infection pattern characterized by a higher incidence of peritonitis [36]. The microbial profile of IE in PD patients mirrors the organisms commonly found in dialysis-associated peritonitis, including Gram-positive bacteria such as staphylococci and enterococci alongside Gram-negative strains [36]. Among these, coagulase-negative staphylococci are among the most common culprits of bacteremia in these patients. However, *S. aureus* remains the most common organism isolated in IE among PD patients [3]. Overall, patients undergoing both HD and PD are more prone to IE caused by Staphylococcus Aureus and fungi compared to non-dialysis patients, as indicated in Table 2 [3]. Knowing the pathogens linked to IE in ESRD patients receiving dialysis is essential for choosing the right antimicrobial treatment, implementing infection control measures, and preventing recurring infections, which differ from patients not undergoing dialysis.

## 4. Cardiac Valves Most Commonly Involved in Infective Endocarditis among Dialysis Patients

Among patients undergoing dialysis, IE commonly affects the mitral valve, followed by the aortic valve [6]. The occurrence of mitral valve involvement ranges between 30.9% and 61.7%, while aortic valve involvement ranges from 22.3% to 42.8%, as seen in Table 2. This trend mirrors what is seen in the non-dialysis population, where the mitral and aortic valves are most frequently impacted as well. The unique hemodynamics and endothelial damage associated with chronic kidney disease and dialysis may explain why the mitral valve is often affected in these patients [37]. Other factors related to dialysis that make the mitral and aortic valves more vulnerable to IE include increased left filling pressures and turbulent blood flow across the valves [38]. This heightened mechanical stress could be why HD patients have higher rates of IE affecting both the mitral and aortic valves compared to non-HD patients, as indicated by studies, like those conducted by Pericas et al., Kwon et al., and Tao et al. (Table 2) [2,13,22]. Moreover, multiple procedures involving access, such as creating an AVF or inserting an HD catheter, can introduce pathogens into the bloodstream, predisposing the valves to bacterial adherence and colonization [39]. In addition to the mitral and aortic valves, the tricuspid valve is also commonly affected in IE among ESRD patients on dialysis, although at significantly lower rates than the left side of the heart, ranging from 7.1% to 11.7%, with some cases reaching up to 38.2%, according to Table 2. Although intravenous drug use is less common among dialysis patients, it remains a risk factor for tricuspid valve involvement in IE, as evidenced by findings in studies such as Durante-Mangoni et al., where 11.9% of patients were affected (Table 2).

Dialysis patents exhibit higher rates of native valve endocarditis (NVE) compared to prosthetic valve endocarditis (PVE) when compared to non-dialysis individuals due to various factors inherent to their medical condition. Our analysis shows that NVE rates vary between 69% and 100% among dialysis patients, while PVE is observed in about 5.8% to 23.8% of cases [18] (Table 2). As previously mentioned, dialysis patients face a risk of bloodstream infections due to vascular access procedures that create direct pathways for pathogens to enter the bloodstream, resulting in bacteremia and the subsequent development of IE. However, it is the existing comorbidities, such as diabetes and CKD, that make these patients more susceptible to calcific valvular disease, acting as a breeding ground for infection in IE at higher rates than for those not on dialysis [12]. Conversely, the occurrence of PVE may be lower in dialysis patients compared to NVE because this group is less likely to undergo surgery due to their high perioperative risk and poor long-term survival after open-heart procedures [40]. In general, among ESRD patients receiving dialysis treatment, IE commonly affects the aortic, mitral, and tricuspid valves, with higher instances of NVE than PVE, when compared to patients not on dialysis. Factors linked to dialysis such as infections related to access and existing health conditions contribute to the predilection for valve involvement in this high-risk population.

## 5. Cardiac Imaging of Infective Endocarditis in Dialysis Patients

IE presents unique challenges in dialysis patients, necessitating a comprehensive approach to diagnosis that incorporates specific considerations for this population. Cardiac imaging plays a pivotal role in the diagnostic pathway, offering crucial insights into valvular involvement and infectious complications [41].

### 5.1. Transthoracic Echocardiography (TTE)

Echocardiography is the most frequently used imaging study (with TTE being the first line) for confirming the diagnosis of IE. According to the 2023 Duke-International Society for Cardiovascular Infectious Diseases (ISCVID), echocardiography can help constitute a major imaging criterion by either (1) detecting valvular/leaflet perforation or aneurysm, abscesses, pseudoaneurysms, or intracardiac fistulas, (2) identifying new valvular regurgitation, or (3) isolating new partial dehiscence of the prosthetic valve [42]. More specifically, TTE is the most frequently used imaging study for confirming the diagnosis of IE (100% of patients received it per (Table 2) [43]. However, the sensitivity of TTE may be reduced in patients with abnormal left ventricular geometry, such as those with the LVH commonly seen in dialysis patients [44]. Furthermore, the impact of left ventricular mass (LVM) inaccuracies as well as volume assessments due to fluid fluctuations can further affect the interpretation of TTE findings in dialysis patients [44]. This often leads to difficulties in identifying vegetations or assessing the severity of valvular regurgitation, which are key components in the diagnosis and management of IE. Given the potential for fluid status to affect TTE measurements, it is important to consider these limitations, especially when using TTE in the initial assessment of IE in dialysis patients. Therefore, the guidelines recommend TEE, either when TTE is inconclusive or if there is a high clinical suspicion of IE, as it is less affected by loading conditions and can better visualize vegetations and paravalvular complications [8,43,45]. This is evident in Table 2, as vegetations were identified only using TEE in 32.3% of the cases in Kwon et al. [13].

### 5.2. Transesophageal Echocardiography (TEE)

Although TEE is less affected by loading conditions and is more accurate than TTE, it still presents several challenges in the investigation of IE among dialysis patients. Firstly, the presence of vascular access devices, especially if infected, can be challenging to visualize on both TTE and TEE due to acoustic shadowing and artifacts [46]. Secondly, HD patients often have altered cardiac anatomy, such as valvular calcifications, which may confound the interpretation of TEE findings [37]. Calcified valves may harbor vegetations that are difficult to differentiate from the calcifications themselves. Moreover, the presence of previous valvular damage or abnormalities, such as scarring or myxomatous changes, can mimic or obscure findings of IE on TEE [7,47]. Third, the timing of the TEE relative to the HD session can affect the visualization of vegetations or abscesses. As stated earlier, volume status can influence the echocardiographic windows and the ability to detect subtle findings, less so in TEE than in TTE. Fourth, TEE may not detect initial paravalvular abscesses, particularly when the study is performed early in the patient’s illness, as these may appear only as nonspecific perivalvular thickening and may become more recognizable as they expand over time [45]. Therefore, such abscesses may require repeat imaging to confirm the diagnosis [7]. Figure 2, Figure 3 and Figure 4 are provided as examples of ESRD patients with mitral and aortic valve endocarditis who underwent TEE.

Overall, it is crucial to be aware of the potential for both false-positive and false-negative results and the need for careful interpretation of TEE findings in the context of the patient’s clinical presentation and dialysis status [7,47]. While TEE is superior to TTE in visualizing vegetations and paravalvular complications, TEE may still not be 100% sensitive for the diagnosis of IE, and a negative TEE does not rule out the disease. Therefore, it is important to recognize the need for repeat imaging or alternative imaging modalities in the diagnostic workup of IE among dialysis patients. Although the dialysis patients in the articles reviewed in Table 2 all underwent TTE and/or TEE, the role of repeat or further imaging outside echocardiography was not included in these studies, which should be taken into consideration in future studies.

### 5.3. Computed Tomography (CT)

Updates in the 2023 Duke-ISCVID IE criteria for diagnosis include cardiac computed tomography (CT) as an imaging major criterion, alongside echocardiography, to accurately identify vegetations, perforations, aneurysms, abscesses, pseudoaneurysms, and/or intracardiac fistulas [42]. Cardiac CT may be useful in the workup of IE, guiding medical and potentially surgical management. The use of contrast is generally a contra-indication in patients with advanced kidney disease not yet on dialysis [8,48], and therefore would limit the utility of cardiac CT in this population. For patients with ESRD already on dialysis, contrast-enhanced cardiac CT may be performed in select cases for appropriate indications. Contrast-enhanced cardiac CT is less accurate than TTE and TEE for identifying valvular vegetations but is valuable for detailed anatomical assessment and identification of complications in IE such as paravalvular abscesses, pseudoaneurysms, and fistulas, especially in PVE where echocardiography may be limited by shadowing artifacts [7,43,45]. Additionally, CT angiography can be helpful in preoperative planning to assess coronary artery and aortic anatomy in patients undergoing surgery for IE complications [7,45]. Lastly, extracardiac contrast-enhanced CT is useful for detecting systemic complications such as pulmonary infarcts, mycotic aneurysms, splenic abscesses, and other septic emboli [43]. Minimizing contrast volume and using iso-osmolar contrast agents are recommended to mitigate the risk of contrast-induced nephropathy, further preventing the loss of any remaining nephron function [49].

Although non-contrast CT cannot be used to evaluate for endocarditis changes, it can be useful for detecting noncardiac complications of IE, such as cerebral, pulmonary, and splenic infarcts, and abscesses [7,8,45]. The consensus of the medical literature is that dialysis patients have higher risks of CNS extracardiac complications compared to non-dialysis patients, indicating the importance of extracardiac CT in the diagnostic workup of IE [41]. However, differentiation between extracardiac complications can be challenging without the use of contrast as although non-contrast CT is a safer alternative for detecting extracardiac complications, its utility is limited by the lack of detailed anatomical resolution compared to contrast-enhanced imaging [45]. This becomes rather important, especially in dialysis patients, as extracardiac findings in these patients could be from IE or they could be associated with other complications of dialysis unrelated to IE such as thrombus formation in the presence of vascular catheters [50,51]. As a result, the use of contrast can be crucial to help differentiate between IE-related and dialysis-related findings. In patients already on dialysis, the timing of CT with contrast should be coordinated with dialysis sessions to manage the volume load from intravenous contrast and to remove the contrast media after imaging [52].

### 5.4. Nuclear Imaging

The 2023 Duke-ISCVID IE criteria for diagnosis now also includes 18F-fluorodeoxyglucose positron emission tomography/computed tomography (18F-FDG PET/CT) as an imaging major criterion to help detect abnormal metabolic activity on native valves, prosthetic valves, and/or other prosthetic material [42]. However, in dialysis patients, the diagnostic capability of 18F-FDG PET/CT may be affected by the chronic inflammatory state of ESRD, potentially leading to false-positive results due to increased FDG uptake not related to infection [53]. Additionally, the presence of vascular grafts and devices commonly used in dialysis patients can also show increased FDG uptake, which may be misinterpreted as infection [53,54]. Therefore, the role of 18F-FDG PET/CT may be limited in comparison to the workup of IE in non-dialysis patients.

Although FDG is the most commonly used tracer in PET/CT, other radiotracers can provide additional information crucial to the diagnosis and management of IE. For example, white blood cell (WBC) single photon emission computed tomography (SPECT)/CT is often used to detect infections particularly associated with cardiac implantable electronic devices (CIEDs) and prosthetic valves [48]. This modality provides high specificity for infection through the accumulation of labeled WBCs at the site of infection, visualized on imaging, in order to enhance the diagnosis of IE, especially when other imaging modalities may be inconclusive [48]. However, there is little evidence that specifically addresses the use of WBC SPECT/CT in ESRD patients with IE. Most studies have focused on broader populations or specific subgroups, such as those with CIED infections, and have yet to focus on ESRD or dialysis patients.

### 5.5. Cardiac Magnetic Resonance (CMR) Imaging

CMR is not a routine primary imaging modality in the diagnosis and management of IE in both non-dialysis and dialysis patients. In non-dialysis patients, CMR can evaluate valvular complications, which may not be adequately visualized by echocardiography. CMR can be used to evaluate the extent of myocardial involvement and quantify valvular regurgitation when echocardiographic findings are inconclusive [43]. However, in dialysis patients with a historic risk for nephrogenic systemic fibrosis [55], the use of gadolinium-based contrast agents in CMR must be carefully considered, with an individual risk versus benefit analysis.

In summary, the diagnosis of IE in dialysis patients requires a multifaceted imaging approach, with TTE as the initial screening tool and TEE as an important imaging modality for confirmation, with cardiac and extracardiac CT and nuclear imaging techniques as complimentary imaging modalities for selecting appropriate patients (Figure 5). Future studies focusing on the diagnostic accuracy and clinical utility of these imaging modalities specifically in dialysis patients with IE are warranted to optimize patient management and improve outcomes.

## 6. Management of Infective Endocarditis in Dialysis Patients

Effective management of IE in patients on dialysis requires a comprehensive approach that addresses both medical and surgical interventions. Given the heightened risk of complications in this population, the ACC/AHA Guidelines recommend the assembly of an Endocarditis Team so that a multidisciplinary approach can be implemented to best serve the patient’s clinical needs with the consideration of local epidemiological factors for the optimization of patient outcomes [8]. However, it is crucial to also involve nephrology and/or transplant nephrology in dialysis patients with native or transplanted kidneys in this multidisciplinary IE approach.

### 6.1. Medical Management

The medical management of IE in dialysis patients includes the administration of appropriate antibiotic therapy, which is guided by the identification of the causative organism. Empiric treatment depends on multiple factors including the patient’s past microscopies and susceptibilities, previous antibiotic therapy, comorbidities, place of infection acquirement, and likely underlying cause of IE (i.e., CVC blood stream infection). Empiric antibiotic therapy in dialysis patients typically includes vancomycin and coverage for Gram-negative bacilli, based on the local antibiogram, such as third-generation cephalosporin, carbapenem, or B-lactam/B-lactamase combination, as seen in Table 3.

The standard of care had previously consisted of intravenous antibiotic treatment for the duration of the IE treatment until the results of the POET trial were made available in 2019, showing that up to 20% of IE patients could complete treatment by oral antibiotic therapy after an initial phase of intravenous antibiotic treatment [57]. Interestingly, of the cohort that received oral antibiotic treatment, 10.4% had coexisting renal failure and 7.5% were on dialysis compared to 12.6% and 6.5% in the intravenous treatment cohort, respectively. The noninferior shift in antibiotic treatment from intravenous to oral was consistent across the patients that had renal failure and who were on dialysis as well [57]. Therefore, the recent shift to two phases (inpatient intravenous treatment for 2 weeks followed by outpatient intravenous antibiotics or oral antibiotic treatment for up to 6 weeks) may also be applied to dialysis patients, although they were not included in the studies reviewed in Table 3 [8]. However, just as with non-dialysis patients, it is important that the transition from inpatient to outpatient only occurs once the repeat TEE confirms the absence of local progression and complications [8].

Overall, the duration of antibiotic therapy for non-dialysis and dialysis patients is the same, ranging from 4 to 6 weeks [7]. However, there are notable differences in the antibiotic choice and management approach. Dialysis patients are more likely to receive vancomycin due to the higher prevalence of MRSA than in non-dialysis patients [3]. Although empiric therapy also often includes vancomycin in the non-dialysis population, it is often tailored more quickly based on local antibiograms and patient-specific factors [3]. In terms of definitive therapy, MSSA IE is treated with cefazolin in both populations; however, the dosing in dialysis patients is adjusted appropriately after dialysis [58]. For MRSA, vancomycin or daptomycin is used in both groups, but dialysis patients require more frequent drug level monitoring due to their altered pharmacokinetics [7]. In summary, while the duration of antibiotics is similar, the choice of therapy and management strategies (i.e., monitoring and timing of administration) differ significantly between dialysis and non-dialysis patients.

### 6.2. Surgical Management

The AHA and ESC include new significant valvular regurgitation, persistent infection (bacteremia or fevers lasting longer than 5–7 days after initiation of directed antimicrobial therapy), evidence of valve dehiscence, perforation, rupture, or fistula, large perivalvular abscess, and increasing size of vegetation despite appropriate antimicrobial therapy as indications for surgical intervention in IE [7,8]. Additionally, in the presence of an implanted cardiac electronic device and definite endocarditis, complete removal of the device is indicated [7]. Early surgery may also be considered in patients with native left-sided valve endocarditis who exhibit mobile vegetations greater than 10 mm in length [7].

The American Association for Thoracic Surgery (AATS) Consensus Guidelines also supports surgical intervention in dialysis patients with IE when indicated based on factors such as the patient’s clinical condition, perioperative risk, capacity to overcome the infection, and long-term prognosis [59]. Several scoring systems are routinely employed to predict several complication risks after cardiac surgery, with the EuroSCORE II (European System for Cardiac Operative Risk Evaluation Score) and the STS (Society of Thoracic Surgeons) scores being the most commonly used [60]. Although these scores are not specific to IE, they are widely utilized in clinical practice to assess whether the potential benefits of surgery outweigh the associated risks. While scoring systems tailored specifically for IE exist, their utility is restricted due to performance variability, leading to limited adoption in practice [8]. Most, if not all, of these scoring systems characterize renal failure and dialysis status as major risk factors that affect survivability after surgery, which often precludes dialysis patients from surgery when other comorbidities are included. Unfortunately, these scoring systems calculate risk based solely on patient characteristics and comorbidities, rather than individualized risk of surgical versus medical management among dialysis patients themselves.

Hospital mortality for hemodialysis patients after valve surgery for IE is reported at 13%, with a 5-year survival rate of 20% [61]. This survival rate is lower than the general population, but higher than that of dialysis patients treated with only medical management for IE. In fact, a recent study showed that the odds of in-hospital and 30-day mortality were lower in dialysis patients that were surgically treated compared to dialysis patients that were only medically treated [62]. In addition, much of the available literature compares outcomes of medical versus surgical management in dialysis patients versus non-dialysis patients, rather than among dialysis patients themselves.

### 6.3. Non-Surgical Interventions

More recent investigations have employed interventional cardiology procedures for the management of complications and removal of infected material in dialysis patients. For right-sided IE, minimally invasive procedures such as the AngioVac system have been utilized to debulk vegetations from valves or cardiac implantable electronic devices [63]. The AngioVac system, which is a vacuum-assisted venous drainage system, has shown a high procedural success rate in removing vegetations, thereby reducing the bacterial load and potentially improving outcomes in these critically ill, high-surgical-risk patients [63]. The AngioVac system has also recently been used on the left side of the heart for patients with ESRD and IE. Specifically, the AngioVac system has been utilized for percutaneous transeptal debulking of mitral valve endocarditis in patients who are at high surgical risk or deemed inoperable [64]. This approach involves careful preprocedural planning, including optimal transeptal puncture height and the use of a sentinel cerebral protection device to decrease the risk of procedure-related cerebral embolism [64]. The procedure also employs a venous extracorporeal membrane cannula for reinfusion to avoid complications related to large-bore arterial access [64]. However, the procedural success rates of the AngioVac system in left-sided IE, particularly in patients with dialysis who are at high surgical risk or deemed inoperable, are not yet well defined in the medical literature. The available data primarily consist of case reports and small series, which suggest that while the AngioVac can be used for debulking mitral valve vegetations in these high-risk patients, further studies are needed to establish its efficacy and safety comprehensively.

## 7. Future Directions

Future research into personalized preventive strategies for dialysis patients at risk for IE is essential in order to improve quality of life and outcomes. First, it is crucial to identify high-risk patients through comprehensive risk stratification models. Therefore, predictive algorithms should be developed to incorporate significant prognosis factors such as prolonged catheter use, previous valvular disease, and elevated baseline C-reactive protein (CRP) in dialysis patients [65]. Second, molecular techniques such as polymerase chain reaction (PCR) and metagenomic sequencing (MGS) are currently being investigated for their potential to identify pathogens in cases where cultures are negative, which can provide rapid and accurate identification of causative organisms, crucial for timely and targeted therapy [42]. The 2023 Duke-International Society for Cardiovascular Infectious Diseases (ISCVID) Criteria have incorporated these molecular diagnostics as part of the updated diagnostic criteria for IE. However, these have yet to be studied within the dialysis population specifically and warrant future investigation [42]. Third, immunological biomarkers are also currently being explored such as serum proteomic analyses investigating osteoprotegerin, which has been seen to be elevated in IE patients compared to non-IE patients [66]. Future studies should look into whether osteoprotegerin is even more elevated in dialysis patients with IE compared to non-dialysis IE patients. Fourth, given the high incidence of Staphylococcus aureus infections in dialysis patients, targeted antibiotic prophylaxis could be beneficial. However, this must be balanced with the risk of antibiotic resistance. Future studies should explore the efficacy and safety of prophylactic antibiotic use in high-risk dialysis patients. Fifth, it is monumental to educate patients on the importance of hygiene, proper catheter care, and early symptom reporting in order to empower them to participate actively in their care and reduce infection rates. By focusing on these areas, future studies may lead to the development of personalized preventive strategies that may improve the quality of life and clinical outcomes for dialysis patients either at risk of or with IE.

## 8. Conclusions

IE continues to be a clinical challenge for patients with ESRD undergoing dialysis, with mortality rates significantly higher than those of the general population. The complex interplay of frequent vascular access, comorbid conditions, and increased susceptibility to infections contributes to the heightened risk of IE in this population. Despite advancements in diagnosis and treatment, early recognition and management remain critical to improving outcomes. Ongoing research focusing on enhancing imaging technology, refining preventive strategies, and developing specific surgical risk scores for IE in dialysis patients holds promise for better patient care. By adopting a multidisciplinary approach and leveraging the latest innovations in cardiovascular imaging and interventions, healthcare providers can strive to improve outcomes for dialysis patients affected by IE.

## Figures and Tables

**Figure 1 healthcare-12-01631-f001:**
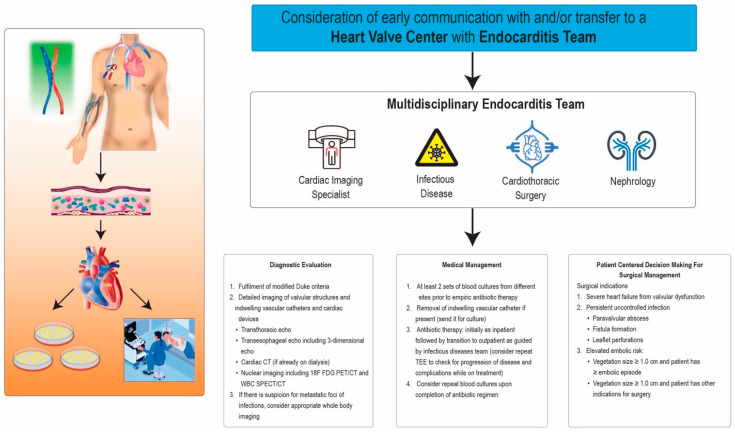
**Central Illustration:** Approach to diagnosis and patient-centered medical and surgical management in dialysis patients with infective endocarditis.

**Figure 2 healthcare-12-01631-f002:**
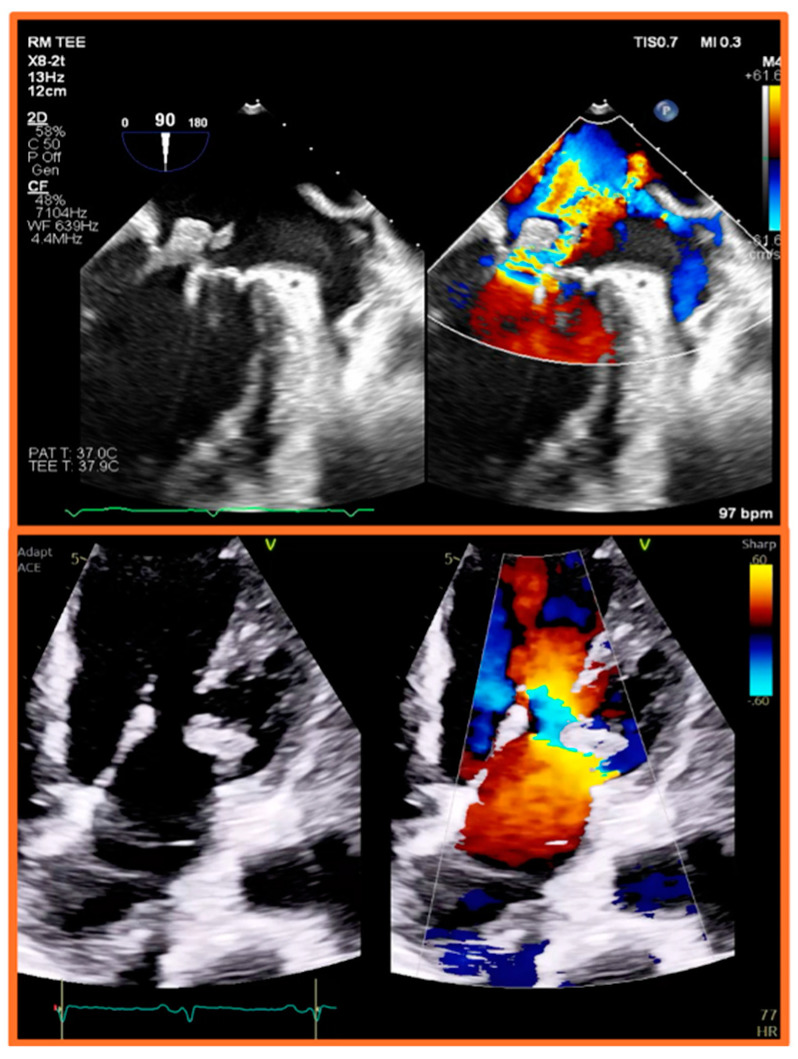
**Mitral valve endocarditis in a patient with end-stage renal disease.** Color-compare images from transesophageal (**top panel**) and transthoracic (**bottom panel**) echocardiograms demonstrating a large mobile vegetation on the posterior leaflet of the mitral valve associated with chordal rupture, leaflet flail, and severe mitral regurgitation. Images obtained from a dialysis patient with infective endocarditis due to Enterococcus Faecalis bacteremia from tunneled-dialysis catheter that had not been removed after arteriovenous fistula (AVF) maturation.

**Figure 3 healthcare-12-01631-f003:**
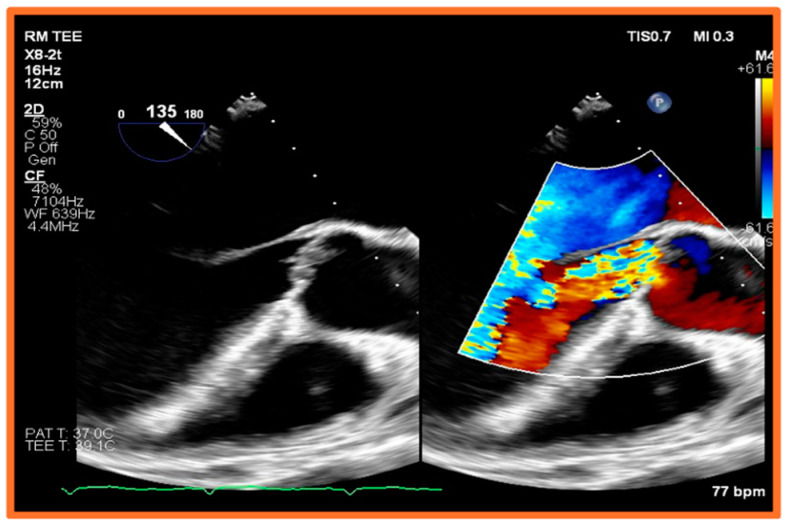
**Aortic valve involvement in a patient with end-stage renal disease and endocarditis.** Color-compare transesophageal echocardiographic image showing a long-axis view of the left ventricle (two-dimensional image shown on the left panel; color Doppler focused on the aortic valve shown on the right panel). The aortic valve leaflets are thickened and exhibit a mobile echodensity, as well as severe aortic regurgitation, most likely due to leaflet perforation. Image is taken from a dialysis patient with infective endocarditis secondary to Enterococcus Faecalis bacteremia from tunneled-dialysis catheter that had not been removed after arteriovenous fistula (AVF) maturation.

**Figure 4 healthcare-12-01631-f004:**
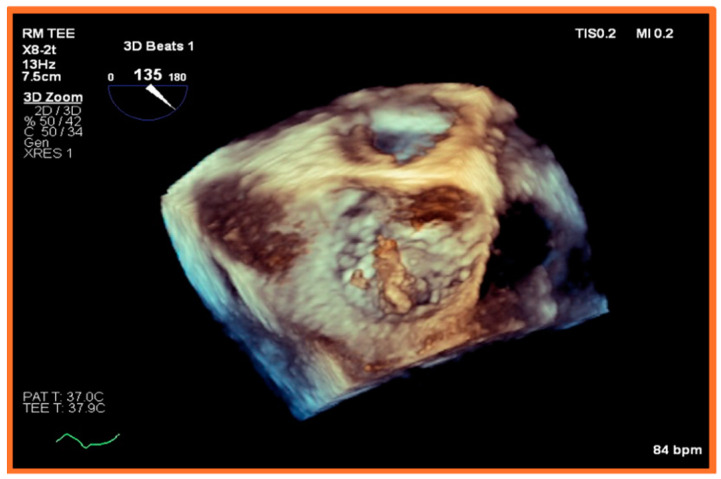
**TEE 3D MV showing large mobile vegetation in a patient with end-stage renal disease and endocarditis.** Transesophageal echocardiographic three-dimensional reconstruction of the mitral valve, view from the left atrium. A large mobile echodensity, consistent with vegetation, is attached to the P2 scallop of the mitral valve and prolapses into the left atrium during systole. Image is taken from a dialysis patient with infective endocarditis secondary to Enterococcus Faecalis bacteremia from tunneled-dialysis catheter that had not been removed after arteriovenous fistula (AVF) maturation.

**Figure 5 healthcare-12-01631-f005:**
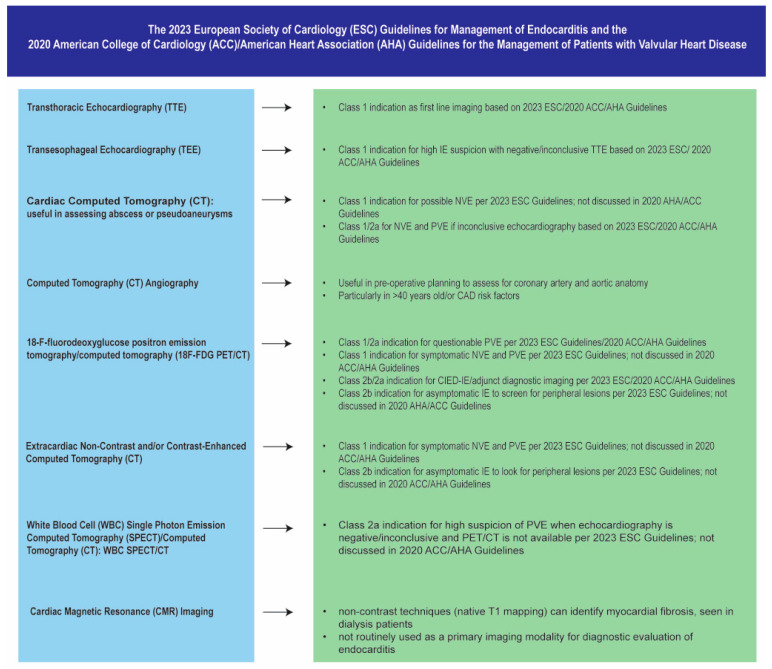
Summary of the 2023 European Society of Cardiology (ESC) Guidelines for Management of Endocarditis and the 2020 American College of Cardiology (ACC)/American Heart Association (AHA) Guidelines for the Management of Patients with Valvular Heart Disease class indications and recommendations for different imaging modalities [8,41,45,55,56].

**Table 2 healthcare-12-01631-t002:** This table provides a comprehensive overview of the 8 reviewed studies between 2010 and 2024, which examined IE in dialysis patients. It includes information on causative organisms, affected valves, valve types, imaging modalities, and imaging findings. Abbreviations: *Staph aureus* (*S. aureus*); Methicillin-resistant Staph Aureus (MRSA); transthoracic echocardiography (TTE); transesophageal echocardiography (TEE); Haemophilus Aggregatibacter, Cardiobacterium, Eikenella, and Kingella (HACEK); Methicillin-sensitive Staphylococcus Aureus (MSSA); Cardiovascular Implantable Electronic Devices (CIEDs). * signifies the data variables which are statistically significantly higher compared to non-hemodialysis patients.

Study	IE Diagnosis Compared to Non-Dialysis Patients	Causative Microorganism(s)	Affected Valve(s)	Valve Type (Native; Prosthetic)	Imaging	Imaging Findings
Zolfaghari et al., 2024 [17] Iran	Acute: 20/34 (58.8%) Subacute: 2/34 (5.8%) Chronic: 3/34 (8.82%) *	Coagulase-negative staphylococci 5/34 (14.71%) *S. aureus* 4/34 (11.7%; MRSA 5.8%) Enterococci 3/34 (8.8%) Fungal infection 3/34 (8.8%)	Mitral valve: 14/34 (41.1%) Tricuspid valve: 13/34 (38.2%) * Aortic valve: 12/34 (35.2%) Pulmonic valve: 2/34 (5.8%)	Native valve: 34/34 (100%) Prosthetic valve: 2/34 (5.88%)	TTE/TEE	Most common vegetation sites were left-sided valves Right-sided valves more affected in HD patients Simultaneous right- and left-sided vegetations (native and prosthetic valves) more common in HD patients
Pericas et al., 2021 [2] International	Length of symptoms to presentation <1 Month: 456/498 (91.6%) * in HD vs. 4402/5608 (78.5%) in non-HD	*S. aureus*: 252 (47.8%) * MRSA: 95/252 (37.7%) * Coagulase-negative staphylococci: 78 (14.8%) * Enterococci: 81 (15.4%) * Non-HACEK Gram-negative: 20 (3.8%) Blood culture negative: 28 (5.3%)	Mitral valve: 173 (31.7%) * Aortic valve: 122 (22.3%) * Tricuspid valve: 39 (7.1%) Aortic + mitral Valve: 59 (10.8%) Mitral + tricuspid Valve: 39 (7.1%) *	Native valve: 419 (77.2%) * Prosthetic valve: 76 (14%) *	TTE/TEE	Intracardiac vegetations: 481 (87.6%) *
Gallacher et al., 2021 [20] Scotland	Not described	*S. aureus*: 21/81 (25.9%) * Coagulase-negative *Staphylococcus* spp.: 8/81 (9.9%) *Streptococcus* spp.: 8/81 (9.9%) Other (polymicrobial/*Enterococcus* spp.): 10/81 (12.3%) Negative blood cultures/blood cultures not taken: 34/81 (42%)	Not described	Not described	Not described	Not described
Kwon et al., 2021 [13] Korea	Definite IE HD: 30/34 (88.2%); non-HD 38/46 (82.6%)	*S. aureus*: 21/34 (61.8%) MSSA: 2/34 (5.9%) MRSA: 19/34 (55.9%) Coagulase-negative *Staphylococcus* spp.: 1/31 (2.9%) *Viridans streptococci*: 2/34 (5.8%) Culture negative: 4/34 (11.8%)	Aortic valve: 8/34 (23.5%) Mitral valve: 21/34 (61.7%) Tricuspid valve: 4/34 (11.7%) Two or more affected valves: 1/34 (2.9%)	Native valve: 34/34 (100%) Prosthetic valve: 0/34 (0%)	TTE: 34/34 (100%) TEE: 25/34 (73.5%)	Identified vegetation: only in TEE 11/34 (32.3%) in HD 14/34 (30.4%) in non-HD *
Bhatia et al., 2017 [3] United States	Not described	*S. aureus*: 18,860/30,995 (60.8%) * Non-aureus staphylococcus: 4541/30,995 (14.7%) * Gram-negative bacilli: 3163/30,995 (10.2%) Fungi: 131/30,995 (0.4%) Streptococcus: 4354/30,995 (14%) Enterococcus: 2734/30,995 (8.8%)	Not described	Not described	Not described	Not described
Durante-Mangoni et al., 2016 [18] Italy	TTE evidence of IE: 34/42 (81%) TEE evidence of IE: 24/28 (85.7%)	*S. aureus*: 15/42 (44.1%) * Staphylococcus: 8/42 (23.5%) Streptococcus: 3/42 (8.8%) Enterococcus: 6/42 (17.7%) Polymicrobial IE: 3/42 (8.8%)	Aortic valve: 13/42 (35%) Mitral valve: 14/42 (35%) Tricuspid/pulmonary valve: 7/42 (15%)	Native valve: 29/42 (69%) Prosthetic valve: 10/42 (23.8%) CIED: 4/42 (7.1%)	TTE: 42/42 (100%) TEE: 28/42 (66.7%)	Valvular regurgitation: Aortic: 15/42 (35.7%) Mitral: 15/42 (35.7%) Tricuspid: 4/42 (9.5%) Pulmonary: 1/42 (2.4%) None: 7/42 (16.7%)
Jones et al., 2013 [21] United Kingdom	100% dialysis patients	*S. aureus*: 17/42 (40.4%) MRSA: 2/42 (4.7%) *S. epidermidis*: 5/42 (11.9%) Culture negative: 5/42 (11.9%) *E. coli*: 1/42 (2.3%) Enterococcus 5/42 (11.9%) Coliforms: 1/42 (2.3%) Unknown: 2/42 (4.76%)	Mitral valve: 13/42 (30.9%) Aortic valve: 18/42 (42.8%) Tricuspid valve: 5/42 (11.9%) 2+ valves: 4/42 (9.5%) Eustachian valve: 1/42 (2.3%) CIED: 1/42 (2.3%)	Native valve: 38/42 (90.4%) Prosthetic valve: 4/42 (9.52%)	TTE: 42/42 (100%)	TTE-identified vegetations: 10/42 (23.8%)
Tao et al., 2010 [22] China	6/6 (100%)	*Klebsiella pneumoniae*: 1/6 (16.6%) Culture negative: 2/6 (33.3%) Enterococcus: 1/6 (16.6%) MRSA: 2/6 (33.3%)	Aortic valve: 2/6 (33.3%) Mitral valve: 3/6 (50%) Mitral + aortic: 1/6 (16.6%)	Native valve: 6/6 (100%)	TTE: 6/6 (100%) TEE: 1/6 (16.6%)	TTE-identified vegetations: 5/6 (83.3%) TEE-identified vegetations: 1/6 (16.6%)

**Table 3 healthcare-12-01631-t003:** This table provides a comprehensive overview of the 8 reviewed studies between 2010 and 2024, which examined IE in dialysis patients. It includes information on medical versus surgical management, antibiotic regimens, extracardiac complications, and in-hospital mortality. Abbreviations: congestive heart failure (CHF); Central Nervous System (CNS); Mitral Valve Replacement (MVR); Aortic Valve Replacement (AVR); Tricuspid Valve Replacement (TVR); Pulmonic Valve Replacement (PVR); Methicillin-resistant Staphylococcus Aureus (MRSA); Acute Decompensated Heart Failure (ADHF).

Study	Medical Management	Antibiotic Regimen	Surgical Management	Surgical Intervention	Extra-Valvular Complications	In-Hospital Mortality	Most Common Cause of Death
Zolfaghari et al., 2024 [17] Iran	Total: 16/34 (47.1%) Survived: 9/16 (56.3%) Dead: 7/16 (43.8%)	Vancomycin (85.2%) Meropenem (55.8%) Ciprofloxacin (29.4%) Linezolid (17.6%) Ceftriaxone (17.65%)	Total: 18/34 (52.9%) Survived: 11/18 (61.1%) Dead: 7/18 (38.9%)	Valve Surgeries	Pulmonary emboli: (29.4%) Lung abscesses: (11.7%) Cerebral emboli: (5.8%) Splenic abscesses: (5.8%) Brain abscesses: (2.9%)	14/34 (41.18%)	Sepsis-induced multi-organ failure
Pericas et al., 2021 [2] International	381/549 (69.3%)	Beta-lactams: 69 (19.7%) Glycopeptides: 82 (23.4%) Aminoglycosides: 14 (4%) Beta-lactams + aminoglycosides: 52 (14.8%) Glycopeptides + aminoglycosides: 63 (17.9%) Glycopeptides + beta-lactams: 16 (4.6%) Glycopeptides + beta-lactams + aminoglycosides: 23 (6.6%)	168/549 (30.6%)	Valve Surgeries	Stroke: 95 (17.2%) Systemic embolization (non-stroke): 104 (18.8%) CHF: 143 (25.9%) Persistent bacteremia: 124 (22.4%) Paravalvular complications: abscess: 72 (13.1%) perforation: 60 (10.9%) Intracardiac fistula: 12 (2.2%) New moderate or severe regurgitation: 288 (52.5%)	168/553 (30.4%)	Not described
Gallacher et al., 2021 [20] Scotland	Not Described	Not described	30/216 (13.9%)	Valve Surgeries	Stroke: 14 (6.5%) Acute heart failure: 21 (9.7%)	40 (18.5%)	Not described
Kwon et al., 2021 [13] Korea	23/34 (67.6%)	Not described	11/34 (32.3%)	Valve Surgeries	Heart failure: 10/34 (29.4%) Conduction abnormality: 3/34 (8.8%) CNS emboli: 7/34 (20.5%) Septic lung: 9/34 (26.4%) Other emboli: 6/34 (17.6%) Stroke: 7/34 (20.5%)	17/34 (50%)	Not described
Bhatia et al., 2017 [3] United States	41,555/44,816 (92.7%)	Not described	Total Valve Replacement surgery: 3261/44,816 (7.3%) AVR: 1802/44,816 (55.3%) MVR: 1857/44,816 (56.9%) TVR: 121/44,816 (3.7%) PVR: 10/44,816 (0.3%) Multiple Valve Replacement: 524/44,816 (16.1%)	Valve Surgeries	Acute stroke: 4480/44,816 (10%)	7495/29,814 (25.1%)	Not described
Durante-Mangoni et al., 2016 [18] Italy	26/42 (61.9%)	Glycopeptides: (35.7 vs. 16.7%) Penicillin G/ampicillin: (21.4 vs. 46.8%)	16/42 (38%) HD 46/126 (36.5%) non-HD Main indications: heart failure and high embolic risk	Mechanical: 62 vs. 65.1% Bioprosthetic: 35.7 vs. 28.6%	Heart failure: 17/42 (40.5%) Intracardiac abscess: 12/42 (9.5%) Stroke: 6/42 (14.3%) Non-cerebral embolism: 16/42 (38.1%) Perforation: 2/42 (4.8%) New valve regurgitation: 9/42 (21.4%) Prosthesis dehiscence: 4/42 (9.5%)	26.2% HD 15.9% non-HD	Older age
Jones et al., 2013 [21] United Kingdom	33/4 (78.5%)	Not described	9/42 (21.4%) Patients receiving surgery were younger and less likely to be infected with *S. aureus*. Average EuroSCORE was 10 for surgically treated patients and 14 for medically treated patients	Valve Surgeries	Extracardiac septic emboli (paraspinal, splenic, colonic, distal limb): 5/42 (12%) Recurrence of endocarditis: 4/42 (9.5%). 1/42 had an intracardiac device that needed replacement 1/42 had prosthetic AV endocarditis that needed replacement	14.3% during initial hospitalization 29.2% at 30 days 32.7% at 3 months	Age > 60 years, septic emboli, and MRSA
Tao et al., 2010 China [22]	3/6 (50%)	Treated for 4 weeks in 2/6 (33.3%) Treated for 6 weeks in 1/6 (16.6%) Treated for 7 weeks in 1/6 (16.6%)	2/6 (33.3%)	Valve Surgeries: 2/6 (33.3%)	Not described	0/6 (0%)	Death due to missed dialysis 2/2 ADHF

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
