# Peer review of "Infective Endocarditis in Patients with End-Stage Renal Disease on Dialysis: Epidemiology, Risk Factors, Diagnostic Challenges, and Management Approaches"

_healthcare, 2024, doi:10.3390/healthcare12161631_

Round 1

Reviewer 1 Report

Comments and Suggestions for Authors

The number of patients with end-stage renal failure, who require hemodialysis, increases annually. In this cohort infective endocarditis (IE) occurs more frequently, then in general population. The disease is characterized by severe complications and high mortality, and pose a number of new challenges for clinicians. The presented review discusses and analyzes in detail the data concerning risk factors, diagnostics, and management of IE. Since in the last 10 years there have been no reviews published on this actual issue, the article will be of interest to readers. The review is well-organized, well written, and analysis all appropriate and relevant studies. However, I have a few suggestions that would help to make the article even more interesting.

1.     Since the title of the article mentions the «epidemiology of IE», it is necessary to add information on the incidence of infective endocarditis among hemodialysis patients.

2.     It is desirable to present data on the clinical features of IE in ESRD patients.

3.     The tables are too cumbersome and difficult to understand. Please consider providing a graphical summary for main table’s columns. For example, you can show on a diagram the frequencies of affected valves in ESRD patients (Mitral (%) Tricuspid (%), Aortic (%) Pulmonic: (%)), based on the published data.

Minor comment

1.     All abbreviations in tables must be explained.

Author Response

Reviewer 1

The number of patients with end-stage renal failure, who require hemodialysis, increases annually. In this cohort infective endocarditis (IE) occurs more frequently, then in general population. The disease is characterized by severe complications and high mortality, and pose a number of new challenges for clinicians. The presented review discusses and analyzes in detail the data concerning risk factors, diagnostics, and management of IE. Since in the last 10 years there have been no reviews published on this actual issue, the article will be of interest to readers. The review is well-organized, well written, and analysis all appropriate and relevant studies. However, I have a few suggestions that would help to make the article even more interesting.

  1. Since the title of the article mentions the «epidemiology of IE», it is necessary to add information on the incidence of infective endocarditis among hemodialysis patients.

Thank you for your suggestion, we have added the epidemiology of IE, including the incidence of IE in dialysis patients to page 4 in the introduction.

“The incidence of IE among dialysis patients, particularly those on hemodialysis, is significantly higher compared to patients not on dialysis and has been increasing in recent years. According to a study from the Nationwide Inpatient Sample database between 2006 and 2011, the hospitalization rate of bacterial endocarditis in dialysis patients increased from 175 to 222 per 10,000 patients within the United States, indicating a rising trend in the incidence of IE among dialysis patients (3). In comparison, the incidence of IE in the general population is significantly lower, around 0.18 per 1000 persons per year, with an estimated incidence rate ratio of 38.1 (4).”

  1. It is desirable to present data on the clinical features of IE in ESRD patients.

Thank you for your suggestion, we have added information on the clinical features of IE in dialysis patients on page 4.

“Clinical manifestations of IE present in dialysis patients include fever, chills, and signs of systemic infections, as seen in non-dialysis patients (6)

  1. The tables are too cumbersome and difficult to understand. Please consider providing a graphical summary for main table’s columns. For example, you can show on a diagram the frequencies of affected valves in ESRD patients (Mitral (%) Tricuspid (%), Aortic (%) Pulmonic: (%)), based on the published data.

Thank you for your suggestion. We found this to be difficult to accomplish given the wide variety of information presented in each reviewed study as well as the large differences in sample sizes and study set-ups of each. This also would not have been able to be accomplished given all of the different columns in each of the 3 tables. We did make the tables less cumbersome.

  1. All abbreviations in tables must be explained.

Thank you for this comment, we have added all abbreviations in the tables in the legends for clarity.

Reviewer 2 Report

Comments and Suggestions for Authors

Thank you for the opportunity to review this manuscript. The topic is interesting from both the scientific and practical points of view. I have a number of comments:

1) Despite the fact that this is a literature review, the authors used the principles of a systematic review. It would be useful to indicate how the publications were selected for analysis. This question arises because the authors selected 8 articles for analysis, including studies conducted on incompatible samples from 6 to 44,816 dialysis patients. It is very difficult to draw correct conclusions in such conditions.

2) The authors should be more precise when presenting the results of other studies. For example, Table 2 indicates that in the study of Zolfaghari et. al. 2024 26.47% of patients had chronic infective endocarditis, while in the article of these authors it is indicated that only 8.82% of patients had this form of the disease. The authors should check the results of their study. 3) In the Medical Management chapter, it would be interesting to see information on the duration of the course of antibacterial drugs in hemodialysis patients with infective endocarditis, as well as on the choice of individual antibiotics or their combinations with justification.

Author Response

Thank you for the opportunity to review this manuscript. The topic is interesting from both the scientific and practical points of view. I have a number of comments:

  • Despite the fact that this is a literature review, the authors used the principles of a systematic review. It would be useful to indicate how the publications were selected for analysis. This question arises because the authors selected 8 articles for analysis, including studies conducted on incompatible samples from 6 to 44,816 dialysis patients. It is very difficult to draw correct conclusions in such conditions.

Thank you for your suggestion. We wanted to select the most current prospective/retrospective studies published over the recent 15 years to provide a more up-to-date review on what studies are available in the literature. We have added our rationale, which can be seen on pages 4-5.

“The goal of this review is to highlight the multifactorial nature of IE risk in dialysis patients, emphasize the roles of vascular access type, dialysis modality, and comorbid conditions in this vulnerable patient population, explore the diagnostic utility of different imaging modalities and the importance of a multidisciplinary approach in managing IE, including both medical and surgical interventions, while providing a focused literature review of the most current prospective/retrospective studies in the last 15 years.”

  • The authors should be more precise when presenting the results of other studies. For example, Table 2 indicates that in the study of Zolfaghari et. al. 2024 26.47% of patients had chronic infective endocarditis, while in the article of these authors it is indicated that only 8.82% of patients had this form of the disease. The authors should check the results of their study.

This was an oversight. Thank you for the correction. The tables have now been corrected and updated.

See page 31: “chronic 3/34 (8.82%)”

  • In the Medical Management chapter, it would be interesting to see information on the duration of the course of antibacterial drugs in hemodialysis patients with infective endocarditis, as well as on the choice of individual antibiotics or their combinations with justification.

That is an excellent point. It would enhance the review to add the duration of the course length and choice of antibiotics in HD patients with IE. You can see this addition on page 19 of the updated manuscript.

“Overall, duration of antibiotic therapy for non-dialysis and dialysis patients is the same, ranging from 4 to 6 weeks (7). However, there are notable differences in antibiotic choice and management approach. Dialysis patients are more likely to receive vancomycin due to the higher prevalence of MRSA than in non-dialysis patients (3).  Although empiric therapy also often includes vancomycin in the non-dialysis population, it is often tailored more quickly based on local antibiograms and patient-specific factors (3). In terms of definitive therapy, MSSA IE is treated with cefazolin in both populations, however the dosing in dialysis patients is adjusted appropriately after dialysis (56). For MRSA, vancomycin or daptomycin is used in both groups, but dialysis patients require more frequent drug level monitoring due to their altered pharmacokinetics (7). In summary, while the duration of antibiotics is similar, the choice of therapy and management strategies (i.e. monitoring and timing of administration) differ significantly between dialysis and non-dialysis patients.”

Reviewer 3 Report

Comments and Suggestions for Authors

I would appreciate the authors the paper provides a thorough review of the epidemiology, risk factors, diagnostic challenges, and management approaches for Infective Endocarditis (IE) in patients with End-Stage Renal Disease (ESRD) on dialysis by including various studies and the analysis of different dialysis modalities are commendable. But I think some sections, particularly the tables, could benefit from clearer presentation and consistency in data reporting. For instance, the data in Table 1 could be better organized with clearer demarcations for variables and ensure consistent use of terms and abbreviations. 

The authors discussed different imaging types on IE but I found lack of discussion on CMR to Dx IE . I would suggest to include a bit discussion on CMR in this context .

As for the diagnostic and management , I would suggest the authors to include one/two figures for diagnostic flow diagram (including different imaging tools) for IE with level of evidence on this imaging since both ACC/AHA and ESC guidelines clearly shown that evidence-based recommendation recently .  

I am also curious with that ; are there any new diagnostic tools or biomarkers under investigation that could improve early detection? 

We know HD patients exhibited a distinct clinical profile of IE with worse short-term outcomes, including higher mortality. As for future direction for prevention and treatment , what can you suggest research/studies into personalized preventive strategies to improve quailty of life and outcomes based on individual risk profiles like ESRD ? 

Author Response

Reviewer 3

I would appreciate the authors the paper provides a thorough review of the epidemiology, risk factors, diagnostic challenges, and management approaches for Infective Endocarditis (IE) in patients with End-Stage Renal Disease (ESRD) on dialysis by including various studies and the analysis of different dialysis modalities are commendable.

  1. I think some sections, particularly the tables, could benefit from clearer presentation and consistency in data reporting. For instance, the data in Table 1 could be better organized with clearer demarcations for variables and ensure consistent use of terms and abbreviations.

Thank you for your suggestion. We have simplified the tables while at the same time retaining the key information.

  1. The authors discussed different imaging types on IE, but I found lack of discussion on CMR to Dx IE. I would suggest including a bit discussion on CMR in this context.

We had initially not included CMR into the discussion as the guidelines have not explicitly stated a role for CMR in IE. However, we provided a CMR section that comments on the potential pros to incorporating CMR into the diagnostic pathway of a dialysis patients with IE, as seen on page 17.

“Cardiac Magnetic Resonance (CMR) Imaging: CMR is not a routine primary imaging modality in the diagnosis and management of IE in both non-dialysis and dialysis patients. In non-dialysis patients, CMR can evaluate valvular complications, which may not be adequately visualized by echocardiography. CMR can be used to evaluate the extent of myocardial involvement and quantify valvular regurgitation when echocardiographic findings are inconclusive. (42) However, in dialysis patients, the use of gadolinium-based contrast agents in CMR must be carefully considered, with an individual risk versus benefit analysis, with a historic risk for nephrogenic systemic fibrosis (54).”

  1. As for the diagnostic and management, I would suggest the authors to include one/two figures for diagnostic flow diagram (including different imaging tools) for IE with level of evidence on this imaging since both ACC/AHA and ESC guidelines clearly shown that evidence-based recommendation recently.

This is a great suggestion. We have created another figure that is a diagnostic flow diagram for IE with the level of evidence of imaging according to the guidelines. Please see Figure 4 at the end of the manuscript.

Summary of the 2023 European Society of Cardiology (ESC) Guidelines for Management of Endocarditis and the 2020 American College of Cardiology (ACC)/American Heart Association (AHA) Guidelines for the Management of Patients with Valvular Heart Disease class indications and recommendations for different imaging modalities (8,40,44,54,66).

  1. I am also curious with that; are there any new diagnostic tools or biomarkers under investigation that could improve early detection?

This is an excellent point. We have added this section into the future directions section created (see #5 for more detail). This can be seen on page 22-23.

“Molecular techniques such as polymerase chain reaction (PCR) and metagenomic sequencing (MGS) are currently being investigated for their potential to identify pathogens in cases where cultures are negative, which can provide rapid and accurate identification of causative organisms, crucial for timely and targeted therapy (40). The 2023 Duke-International Society for Cardiovascular Infectious Diseases (ISCVID) Criteria have incorporated these molecular diagnostics as part of the updated diagnostic criteria for IE, but these have yet to be studied within the dialysis population specifically (40). Immunological biomarkers are also currently being explored such as serum proteomic analyses investigating osteoprotegerin, which has been seen to be elevated in IE patients compared to non-IE patients, however this has not been studied in dialysis patients of yet (41).”

  1. We know HD patients exhibited a distinct clinical profile of IE with worse short-term outcomes, including higher mortality. As for future direction for prevention and treatment, what can you suggest research/studies into personalized preventive strategies to improve quality of life and outcomes based on individual risk profiles like ESRD?

We appreciate your suggestions and have added a future directions section prior to the conclusion section. This can be seen on pages 22-23.

“Future research into personalized preventive strategies for dialysis patients at risk for IE is essential in order to improve quality of life and outcomes. First, it is crucial to identify high-risk patients through comprehensive risk stratification models. Therefore, predictive algorithms should be developed to incorporate significant prognosis factors such as prolonged catheter use, previous valvular disease, and elevated baseline C-reactive protein (CRP) in dialysis patients (63).  Second, molecular techniques such as polymerase chain reaction (PCR) and metagenomic sequencing (MGS) are currently being investigated for their potential to identify pathogens in cases where cultures are negative, which can provide rapid and accurate identification of causative organisms, crucial for timely and targeted therapy (41). The 2023 Duke-International Society for Cardiovascular Infectious Diseases (ISCVID) Criteria have incorporated these molecular diagnostics as part of the updated diagnostic criteria for IE. However, these have yet to be studied within the dialysis population specifically and warrant future investigation (41). Third, immunological biomarkers are also currently being explored such as serum proteomic analyses investigating osteoprotegerin, which has been seen to be elevated in IE patients compared to non-IE patients (64). Future studies should look into whether osteoprotegerin are even more elevated in dialysis patients with IE compared to non-dialysis IE patients. Fourth, given the high incidence of Staphylococcus aureus infections in dialysis patients, targeted antibiotic prophylaxis could be beneficial. However, this must be balanced with the risk of antibiotic resistance. Future studies should explore the efficacy and safety of prophylactic antibiotic use in high-risk dialysis patients. Fifth, it is monumental to educate patients on the importance of hygiene, proper catheter care, and early symptom reporting in order to empower them to participate actively in their care and reduce infection rates. By focusing on these areas, future studies may lead to the development of personalized preventive strategies that may improve the quality of life and clinical outcomes for dialysis patients either at risk of or with IE.”

Round 2

Reviewer 2 Report

Comments and Suggestions for Authors

I am satisfied with the corrections made by the authors. I propose to accept the article for publication as it is.

Reviewer 3 Report

Comments and Suggestions for Authors

Thank you for addressing the feedback and revising the manuscript accordingly. The changes you've made have significantly improved the clarity and depth of your work. The additional data and explanations on IE with ESRD provided are both relevant and well-integrated into the manuscript. I believe this version is much stronger and will contribute meaningfully to the field .